# Three-Dimensional Evaluation on Accuracy of Conventional and Milled Gypsum Models and 3D Printed Photopolymer Models

**DOI:** 10.3390/ma12213499

**Published:** 2019-10-25

**Authors:** Jae-Won Choi, Jong-Ju Ahn, Keunbada Son, Jung-Bo Huh

**Affiliations:** 1Department of Prosthodontics, Dental Research Institute, Institute of Translational Dental Sciences, School of Dentistry, Pusan National University, Yangsan 50612, Korea; won9180@hanmail.net (J.-W.C.); tarov0414@daum.net (J.-J.A.); 2Department of Dental Science, Graduate School, Kyungpook National University, 2177 Dalgubeol-daero, Jung-gu, Daegu 41940, Korea; sonkeunbada@gmail.com

**Keywords:** dental model, trueness, precision, milling, 3D printing

## Abstract

The aim of this study was to evaluate the accuracy of dental models fabricated by conventional, milling, and three-dimensional (3D) printing methods. A reference model with inlay, single crown, and three-unit fixed dental prostheses (FDP) preparations was prepared. Conventional gypsum models (CON) were manufactured from the conventional method. Digital impressions were obtained by intraoral scanner, which were converted into physical models such as milled gypsum models (MIL), stereolithography (SLA), and digital light processing (DLP) 3D printed photopolymer models (S3P and D3P). Models were extracted as standard triangulated language (STL) data by reference scanner. All STL data were superimposed by 3D analysis software and quantitative and qualitative analysis was performed using root mean square (RMS) values and color difference map. Statistical analyses were performed using the Kruskal–Wallis test and Mann–Whitney U test with Bonferroni’s correction. For full arch, the RMS value of trueness and precision in CON was significantly smaller than in the other groups (*p* < 0.05/6 = 0.008), and there was no significant difference between S3P and D3P (*p* > 0.05/6 = 0.008). On the other hand, the RMS value of trueness in CON was significantly smaller than in the other groups for all prepared teeth (*p* < 0.05/6 = 0.008), and there was no significant difference between MIL and S3P (*p* > 0.05/6 = 0.008). In conclusion, conventional gypsum models showed better accuracy than digitally milled and 3D printed models.

## 1. Introduction

Accurate impressions and working models are important elements for fabricating precise dental prostheses [1]. The traditional method is to obtain an intraoral impression by using an elastomeric impression material and fabricate a gypsum cast [2]. However, this procedure causes patient discomfort such as pain, gagging, and inconvenient taste, and the accuracy of the impression is also greatly influenced by the skill of the operator [3]. In addition, disadvantages such as deformation of impression material or model material, contamination by saliva, and blood in the oral cavity have been reported [4,5].

To overcome this drawback, a three-dimensional (3D) digital impression and model using an intraoral scanner (IOS) may be an alternative [6]. This method eliminates steps such as the selection of trays, the polymerization of impression material or the transport to a dental laboratory [7]. In addition, data can be stored permanently, require little storage space, and can be transferred digitally [8]. However, physical models are still needed in diagnostics, appliance manufacturing, manual veneering restorations, removable restorations, and complex prosthetic treatments [6,9,10].

Virtual dental casts from the digital impression can be fabricated into physical models by subtractive and additive manufacturing [11]. Subtractive manufacturing cuts or mills blocks or disks into 3D shapes, whereas additive manufacturing builds 3D shapes by accumulating material layer by layer [6,12].

On the other hand, adaptation of the final prosthesis may be affected by dimensional inaccuracies of dental casts [13]. The accuracy of the dental model is described by trueness and precision (ISO 5725-1). Trueness indicates how far the experimental value is from the reference value [14,15]. Higher trueness indicates that the experimental value is very similar or equivalent to the reference value [15]. Precision indicates how close the repeat measurements in the test group are to each other [16].

Previous studies evaluating the accuracy of dental models manufactured by subtractive manufacturing or additive manufacturing have mainly been limited to diagnostic models in the field of orthodontics [17,18,19]. In the field of prosthetics, it has been limited to single teeth [10,20] or partial mouths [21,22,23], or has limited fabrication methods [9,24,25]. Meanwhile, gypsum blocks for milling have recently been developed. According to the manufacturer, the newly developed gypsum block is cheaper than conventional polyurethane block or polymethyl methacrylate (PMMA) block and has no elasticity; thus, the digital model can be reproduced more accurately. Therefore, the purpose of the present study is to compare the accuracy (trueness and precision) of full arch and prepared teeth of the conventional gypsum model, digitally milled gypsum model, and two digitally 3D printed photopolymer models.

## 2. Materials and Methods

For this experiment, maxillary typodont model (PER5001-UL-SCP-AK-28, Nissin Dental, Kyoto, Japan) including complete dentition and four prepared teeth (#16, #11, #24, and #26) was used as a reference model. In the experiment, #16 used Class II mesio-occlusal (MO) cavity ready-made teeth for inlay preparation and #11 used jacket crown ready-made teeth for single crown preparation, whereas #24 and #26 used full cast crown ready-made teeth for three-unit fixed dental prostheses (FDP) preparation.

Figure 1 shows the overall overview of this study. According to the fabrication method, the model was classified into four groups as follows (each *n* = 10). (1) Conventional gypsum model (CON): Impressions were taken using custom trays fabricated with self-polymerizing resin (Vertex Trayplast NF, Vertex Dental BV, Zeist, Netherlands) and silicone impression materials (Imprint II VPS Impression Material, 3M ESPE, St Paul, MN, USA) according to conventional methods. Type III dental stone (Die stone, DK Mungyo Co., Gimhae, Korea) was mixed according to the manufacturer’s recommended water/powder (W/P) ratio and poured into the impression bodies. The gypsum models were allowed to set for 40 minutes before removal from the impression bodies (Figure 2a). (2) Milled gypsum model (MIL): To reduce light reflection during the scanning processes, powder (EASY SCAN, PD Dental, Seoul, Korea) was applied to the reference model, and the reference model was scanned with an intraoral scanner (TRIOS 3, 3Shape, Copenhagen, Denmark) to obtain 3D virtual model data. These data were converted to a physical model by milling gypsum blocks (Gypsum block, Mungyo Co., Gimhae, Korea) with a five-axis milling machine (Cameleon, Neo Biotech, Seoul, Korea) (Figure 2b). (3) Stereolithography (SLA) 3D printed photopolymer model (S3P): 3D virtual model data were fabricated as a physical model using SLA 3D printer (ZENITH U, Dentis, Daegu, Korea) and photopolymer material (ZMD-1000B TEMPORARY, Dentis, Daegu, Korea) with build angle of 180° and layer thickness of 50 µm. The printed models were cleaned with ethanol for 10 minutes and post-cured for 10 minutes in an ultraviolet (UV) curing machine (DIO PROBO Cure, Dentis, DIO, Busan, Korea) (Figure 2c). (4) Digital light processing (DLP) 3D printed photopolymer model (D3P): Physical models of 3D virtual model data acquired by intraoral scanner were fabricated with DLP 3D printer (DIO PROBO, DIO, Busan, Korea) and photopolymer material (3DCNB-50, DIO, Busan, Korea). Build angle, layer thickness, and postprocessing were consistent with S3P (Figure 2d). The reference model and the experimental models fabricated by each method were scanned with a reference scanner (Identica Blue, Medit, Seoul, Korea) with an accuracy of less than 10 µm to obtain scan files in standard triangulated language (STL) format. Meanwhile, 3D printed photopolymer models (S3P and D3P) were coated with powder before scanning to prevent light reflection on the material.

To evaluate the full arch accuracy and the trueness of the prepared teeth, a 3D analysis program (Geomagic Control X, 3D systems, Rock Hill, SC, USA) was used. The trueness of full arch and prepared teeth were analyzed by superimposing STL data of reference and experimental models through initial alignment and best-fit alignment (each *n* = 10). On the other hand, full arch precision was measured by overlapping STL data of each experimental model (each *n* = 45). The root mean square (RMS) value and color difference map were used for quantitative and qualitative analysis between the reference and experimental objects. The RMS value was calculated by Equation (1):(1)$$\mathrm{RMS}=\frac{1}{\sqrt{n}}\times \sqrt{\overset{n}{\underset{i=1}{\sum }}{\left(X_{1,\;i}-X_{2,\;i}\right)}^{2}}$$
where X_1,i_ is the measuring point i on the reference data, X_2,i_ is the measuring point i on the experimental data, and n is the total number of measuring points. Lower RMS values indicate higher 3D agreement of the superimposed data.

For statistical analysis, SPSS software version 25.0 (SPSS Inc., Chicago, IL, USA) was used. Shapiro–Wilk test and Levene test were used to verify variance normality and homogeneity of variance. The Kruskal–Wallis test (α = 0.05), Mann–Whitney U test and Bonferroni correction (α = 0.05/6 = 0.008 or α = 0.05/3 = 0.017) were used to evaluate the significance between and within each group.

## 3. Results

The accuracy of the full arch for the four groups is shown in Table 1. For full arch, the RMS value of trueness in CON was significantly smaller than in the other groups (*p* < 0.05/6 = 0.008), and there was no significant difference between MIL, S3P, and D3P (*p* > 0.05/6 = 0.008). The RMS value of precision in CON was also significantly smaller than that of the other groups (*p* < 0.05/6 = 0.008). On the other hand, the RMS value of MIL was smaller than S3P and D3P (*p* < 0.05/6 = 0.008), and there was no statistically significant difference between S3P and D3P (*p* > 0.05/6 = 0.008).

In the color difference map for the accuracy of the full arch, recorded discrepancies were set in the range of ±50 µm (20 color segments) and acceptable discrepancies were set in the range of ±50 µm (green color) (Figure 3). In the trueness comparison, CON was shown in green for the overall arch, whereas MIL, S3P, and D3P showed uneven deviations for most of the posterior region (occlusal, buccal, and lingual surface). In addition, blue areas were observed in the anterior region in MIL. In the precision comparison, most of the arch in CON and MIL was within the tolerance range (green color). S3P showed positive deviations (yellow color) on the occlusal surface of the overall arch, whereas D3P showed areas of underrepresentation and overrepresentation over the anterior and posterior regions.

The trueness of each group for the three prepared teeth is shown in Table 2. Regardless of the three prepared teeth, the RMS value of CON was significantly smaller than that of other groups (*p* < 0.05/6 = 0.008), and there was no significant difference between MIL and S3P (*p* > 0.05/6 = 0.008). On the other hand, the RMS value of S3P was significantly lower than that of D3P for the remaining prepared teeth except inlay preparation (*p* < 0.05/6 = 0.008).

In the comparison of the three prepared teeth in each group, the RMS value of the inlay preparation was relatively small in all groups except MIL. In addition, the RMS value of three-unit FDP preparation was the highest in all groups except the CON group. On the other hand, in S3P and D3P, the RMS value of the inlay preparation was significantly smaller than that of the three-unit FDP preparation (*p* < 0.05/3 = 0.017).

In the color difference map for the trueness of the prepared teeth, recorded discrepancies were set in the range of ±100 µm (20 color segments) and acceptable discrepancies were set in the range of ±10 µm (green color) (Figure 4). Regardless of the three prepared teeth, CON showed light deviations relative to the other groups. Meanwhile, MIL, S3P, and D3P showed similar deviation patterns in three prepared teeth. For inlay preparation, a distinct dark red area was observed in the MIL, and the MIL and D3P showed relatively larger deviations than the S3P for single crown and three-unit FDP preparations.

## 4. Discussion

Gypsum models made from conventional impressions using elastomeric impression materials have been used as standard of care for diagnosis, treatment planning, and fabrication of prostheses for the past several years [6,15]. However, these models are vulnerable to the risk of loss, destruction, and deterioration [26]. In recent years, computer-aided design/computer-aided manufacturing (CAD/CAM) technology has led to innovative changes in many industries, as well as in dentistry, providing better clinical experience and quality [27,28]. As an example, the 3D virtual model was made from a digital impression through an intraoral scan of the patient’s mouth [29,30]. If necessary, the 3D virtual model can be manufactured as a physical model through milling and 3D printing [29,31]. Meanwhile, a gypsum block has recently been developed as milling material for making a digital impression into a physical model. Unfortunately, studies of this material have not been reported yet.

To date, the accuracy of dental models has been measured primarily by linear distance measurements [32,33,34]. However, this method is limited because of the lack of measuring points and the inability to measure repeatable measuring points [35]. In addition, the drawback is that a clear reference marker with a specific shape for the measurement is needed, and it is impossible to represent 3D changes of the model [32,34]. Recently, 3D analysis software is being increasingly used to analyze model deviations [25,36]. Because of the alignment and superimposition procedures, 3D comparisons are performed by the computer, avoiding artificial errors in manual measurements [25,37]. The analysis fully reflects the deviations of all points in the model in 3D space, allowing for a more comprehensive and stable evaluation [37]. In addition, the deviation range and area of the model can be instinctively confirmed through the color difference map [25,37]. Previous studies have shown that 3D analysis is more reliable and valid than conventional manual measurements [17,38]. Therefore, the present study used 3D analysis to evaluate the accuracy of the model. 

SLA and DLP technology is the most commonly used in dentistry in terms of printing accuracy, speed, cost, quality, and the possibility of printer miniaturization [37], and thus it is the 3D printing method used in this study. SLA technology forms each layer by irradiating a UV laser to the photopolymer along the contour of the object [6]. After the layer is polymerized, the platform moves vertically by the layer thickness, and the laser cures the new layer [6]. By repeating this process, a 3D object is created [39]. DLP technology is very similar to the curing method of SLA technology, but with different light sources [40]. DLP technology cures the entire layer at once by a high-resolution projector [37,40]. On the other hand, the parameters of the present study for the 3D printing process were set to build angle and layer thickness [41]. In order to reduce printing errors, the occlusal plane of the 3D printed model was aligned parallel to the platform and the build angle was set to 180° to locate the tooth cusp away from the platform [18]. In addition, considering the layer thickness value and printing time that can be set in the two 3D printers used in this study, 50 µm was determined as the layer thickness.

For the full arch, CON showed better accuracy than the other groups, which is consistent with previous studies [9,10,20,22,24]. The accuracy of conventional gypsum models is determined by the type and material of the tray and the type of gypsum [42]. In the present study, custom trays were used to increase the reliability of impressions [9], and polyvinyl siloxane impression materials with excellent dimensional stability were used [2]. On the other hand, the low accuracy of digitally fabricated models (MIL, S3P, and D3P) may be related to cumulative errors at each stage of the digital workflow, such as software, scanner, milling, and 3D printing errors [23,43]. Previous studies comparing the accuracy of intraoral scanners for the full arch reported the largest deviations in the posterior area of all scan data [15,36,44]. In addition, transformation errors may occur when transmitting digital data to CAD software [35,45], and inaccuracies may occur depending on the reproducibility of detail parts of 3D object in milling and 3D printing methods [23]. Figure 3 shows the inaccuracies caused by cumulative errors.

MIL showed better precision of the full arch than S3P and D3P, which may be associated with the postprocessing procedure. Because the MIL only needs to remove the sprue after milling the gypsum block, no additional postprocessing is required. On the other hand, S3P and D3P not only need a cleaning process, but also a post-curing process to increase the mechanical properties of the printed model [6]. The less-cured photopolymer material may remain on the printed model surface due to carelessness of the operator during the cleaning process, and shrinkage or warpage may occur because of additional polymerization processes [6]. The aforementioned description can be confirmed by the positive and negative errors seen in the incisal and occlusal surfaces of S3P and D3P color difference maps (Figure 3).

For all prepared teeth, MIL showed a reproducibility of the model similar to S3P. According to the manufacturer, the gypsum block has no porosities due to its high density and heat resistance, and the chipping of the margin area is reduced using the special mixing method. As a result of this experiment, no errors in the material itself such as fracture or chipping were found. However, the deviations shown in Figure 4 may be due to an error caused by the milling machine. That is, the machined surface decreases over time because of the wear of the tools, and the reproducibility of the detailed structure can be limited by the shape and size of the milling bur [46]. Factors such as thermal error in the cutting tool, vibration error in the machine, and tool deflection error can also affect dimensional stability [46]. Cutting angle or direction may be limited by the shape of the prosthesis, the axis direction of the abutment teeth, and the thickness of the block [47].

S3P showed better trueness than D3P for all prepared teeth except inlay preparation, which is consistent with previous study [19]. Compared with DLP technology, which uses a projector to cure one layer at a time, SLA technology uses a laser beam to cure the photopolymer material from point to line and surface [37]. Thus, more complete polymerization is achieved over the overall thickness of the layer [48]. As shown in Figure 4, D3P shows relatively dark colors for all prepared teeth compared to S3P. On the other hand, 3D printed photopolymer models (S3P and D3P) showed excellent trueness in order of inlay, single, and three-unit FDP preparation, which is consistent with previous studies that show that the accuracy of 3D printed models is inferior as the span increases [9].

In this study, the typodont model was used to provide excellent durability, minimize deformation of temperature and moisture, and provide the same conditions [24]. However, the chemical composition, surface structure, and optical properties are different from natural teeth and failed to reproduce clinical situations such as saliva, patient movement, and anatomical features (tongue, lips, and cheeks) [49]. In addition, the present study did not consider factors such as *x–y* resolution, exposure time, manufacturing environments, and used printing materials that can affect the accuracy of 3D printers [37]. The type and performance of the digital scanners and dental 3D printers used may also influence the results of this study [15,24]. Therefore, further research should be conducted considering more various environments, factors, and clinical situations.

## 5. Conclusions

Within the limitations of this in vitro study, the conventional gypsum models showed better accuracy than digitally milled gypsum models and 3D printed photopolymer models for full arch and prepared teeth. Models manufactured by the SLA technique and milling method showed similar trueness in terms of full arch and prepared teeth. On the other hand, the milling method showed better statistically superior precision of the full arch than the 3D printing method.

## Figures and Tables

**Figure 1 materials-12-03499-f001:**
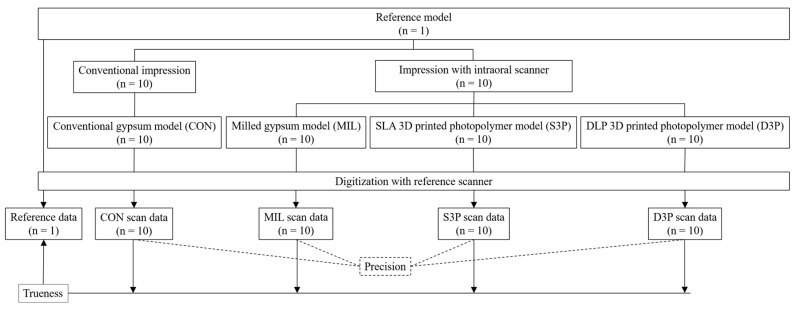
Flow-chart of the study design. SLA, stereolithography; 3D, three-dimensional; DLP, digital light processing.

**Figure 2 materials-12-03499-f002:**
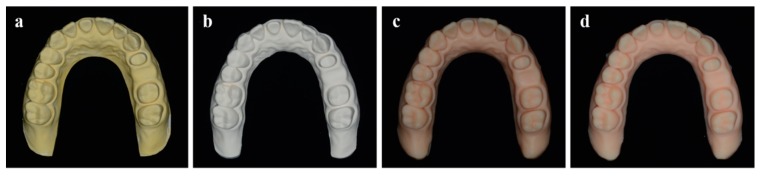
Experimental models. (**a**) Conventional gypsum model (CON); (**b**) milled gypsum model (MIL); (**c**) SLA 3D printed photopolymer model (S3P); (**d**) DLP 3D printed photopolymer model (D3P).

**Figure 3 materials-12-03499-f003:**
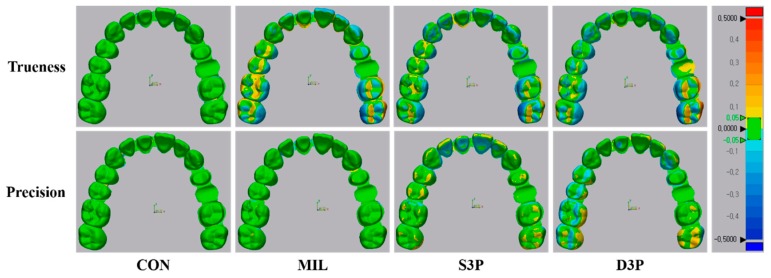
Qualitative analysis of accuracy (trueness and precision) of full arch situation in tested group. CON, conventional gypsum model; MIL, milled gypsum model; S3P, SLA 3D printed photopolymer model; D3P, DLP 3D printed photopolymer model.

**Figure 4 materials-12-03499-f004:**
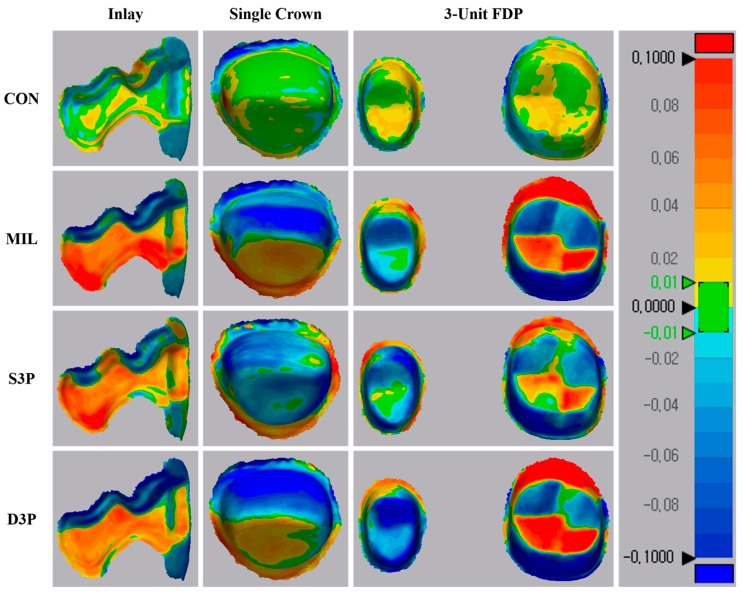
Qualitative analysis of trueness of prepared teeth in tested group. CON, conventional gypsum model; MIL, milled gypsum model; S3P, SLA 3D printed photopolymer model; D3P, DLP 3D printed photopolymer model.

**Table 1 materials-12-03499-t001:** The accuracy (trueness and precision) of four groups for full arch.

Group	Root Mean Square (RMS) (µm)			
	Trueness		Precision	
	Mean ± SD	95% CI	Mean ± SD	95% CI
CON	27.9 ± 2.7^a^	26.0–29.8	20.0 ± 3.1^a^	19.1–21.0
MIL	94.0 ± 11.5^b^	85.8–102.3	37.8 ± 7.0^b^	35.7–39.9
S3P	85.2 ± 13.1^b^	75.9–94.6	49.6 ± 12.1^c^	45.9–53.2
D3P	105.5 ± 22.5^b^	89.4–121.6	52.8 ± 17.5^c^	47.6–58.1
*p*	<0.001		<0.001	

Values followed by the same letter were not significantly different (*p* > 0.05/6 = 0.008). *p*-values are from a Kruskal–Wallis test. RMS, root mean square; SD, standard deviation; CI, confidence interval; CON, conventional gypsum model; MIL, milled gypsum model; S3P, SLA 3D printed photopolymer model; D3P, DLP 3D printed photopolymer model.

**Table 2 materials-12-03499-t002:** The trueness of three prepared teeth in inlay, single crown, and three-unit fixed dental prostheses (FDP).

Group	Root Mean Square (RMS) (µm)			
	Inlay	Single Crown	3-Unit FDP	*p*
	Mean ± SD (95% CI)	Mean ± SD (95% CI)	Mean ± SD (95% CI)	
CON	20.9 ± 1.4^aA^ (19.9–21.9)	30.5 ± 8.4^aB^ (24.5–36.5)	22.2 ± 2.3^aA^ (20.6–23.8)	0.005
MIL	72.8 ± 12.4^bA^ (63.9–81.7)	65.2 ± 6.6^bB^ (60.5–69.9)	95.8 ± 29.1^bcA^ (74.9–116.6)	0.007
S3P	60.7 ± 12.1^bA^ (52.1–69.4)	62.0 ± 10.6^bA^ (53.0–68.0)	81.5 ± 14.1^bB^ (71.4–91.6)	0.004
D3P	66.9 ± 20.9^bA^ (51.9–81.8)	84.3 ± 22.5^cB^ (68.2–100.4)	117.3 ± 38.5^cC^ (89.8–144.8)	0.001
*p*	<0.001	<0.001	<0.001	

Values followed by the same lowercase letter in columns were not significantly different (*p* > 0.05/6 = 0.008). Values followed by the same uppercase letter in rows were not significantly different (*p* > 0.05/3 = 0.017). *p*-values are from a Kruskal–Wallis test. RMS, root mean square; SD, standard deviation; CI, confidence interval; CON, conventional gypsum model; MIL, milled gypsum model; S3P, SLA 3D printed photopolymer model; D3P, DLP 3D printed photopolymer model.

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
