# Peer review of "Three-Dimensional Evaluation on Accuracy of Conventional and Milled Gypsum Models and 3D Printed Photopolymer Models"

_materials, 2019, doi:10.3390/ma12213499_

Round 1
Reviewer 1 Report
This study evaluated the accuracy of dental models fabricated by conventional, milling, and 3D printing methods. By using Statistical analyses with the Kruskal–Wallis test and Mann–Whitney U test with Bonferroni’s correction, it is found the RMS value of trueness and precision in CON was significantly smaller than in the other groups, and there was no significant difference between S3P and D3P. The conclusion of this paper is that conventional gypsum models showed better accuracy than digitally milled and 3D printed models.
While this paper provides authentic experimental data for the accuracy of three different type of dental models, the conclusion is kind of arbitrary. Instead of claiming that CON is much accurate than S3P and D3P in general, it is better to claim that “with the materials and equipment as mention in the paper, the CON is better in trueness and precision”. As far as I know, different brand of digital scanners have quite different performance. Also, the high-end dental 3D printers is much better than the low-end ones. The conclusion part of this paper is an arbitrary decision.
Author Response
This study evaluated the accuracy of dental models fabricated by conventional, milling, and 3D printing methods. By using Statistical analyses with the Kruskal–Wallis test and Mann–Whitney U test with Bonferroni’s correction, it is found the RMS value of trueness and precision in CON was significantly smaller than in the other groups, and there was no significant difference between S3P and D3P. The conclusion of this paper is that conventional gypsum models showed better accuracy than digitally milled and 3D printed models.
Answer) Thank you for your mention. In order to make the sentence more evident, the following correction was made.
Within the limitations of this in vitro study, the conventional gypsum models showed better accuracy than digitally gypsum milled and 3D printed photopolymer models for full arch and prepared teeth.
While this paper provides authentic experimental data for the accuracy of three different type of dental models, the conclusion is kind of arbitrary. Instead of claiming that CON is much accurate than S3P and D3P in general, it is better to claim that “with the materials and equipment as mention in the paper, the CON is better in trueness and precision”. As far as I know, different brand of digital scanners have quite different performance. Also, the high-end dental 3D printers is much better than the low-end ones. The conclusion part of this paper is an arbitrary decision.
Answer) Thank you for your sincere advice. The sentence was added and edited as following.
The type and performance of the digital scanners and dental 3D printers used may also influence the results of this study [15,24].
Reviewer 2 Report
The manuscript by Choi et al evaluated the accuracy of 3D printing with respect to conventional gypsum modeling for the preparation of fixed dental prostheses.
As it is common in other fields of 3D printing application, accuracy in reproducing body models for translational purposes is essential, but still a major and difficult objective to achieve.
The detailed analysis performed in this research by combining sophisticated computational modeling and nonparametric statistical tests is of interest for dental reconstruction applications but also can be translated in other medical fields.
Reviewer 3 Report
Please cite: M. Lin, N. Firoozi, C.-T. Tsai, M. B. Wallace,Y. Kang, 3D-printed flexible polymer stents for potential applications in inoperable esophageal malignancies, Acta biomaterialia, 83 (2019), pp. 119-129. what is the printing parameter?
Author Response
Please cite: M. Lin, N. Firoozi, C.-T. Tsai, M. B. Wallace,Y. Kang, 3D-printed flexible polymer stents for potential applications in inoperable esophageal malignancies, Acta biomaterialia, 83 (2019), pp. 119-129. what is the printing parameter?
Answer) Thank you for your comment. Based on your kind comment, it was modified as below.
On the other hand, the parameters of present study for the 3D printing process were set to build angle and layer thickness [41]. In order to reduce printing errors, the occlusal plane of the 3D printed model was aligned parallel to the platform and the build angle was set to 180 ° to locate the tooth cusp away from the platform [18]. In addition, considering the layer thickness value and printing time that can be set in the two 3D printers used in this study, 50 µm was determined as the layer thickness.
41. Lin, M.; Firoozi, N.; Tsai, C.T.; Wallace, M.B.; Kang, Y. 3D-printed flexible polymer stents for potential applications in inoperable esophageal malignancies. Acta Biomater. 2019, 83, 119-129.